# Demand Forecasting of Online Car-Hailing with Combining LSTM + Attention Approaches

Xiaofei Ye [1], Qiming Ye [1], Xingchen Yan [2], Tao Wang [3,*], Jun Chen [4] and Song Li [1]

1 Ningbo Collaborative Innovation Center for Port Trade Cooperation and Development, School of Maritime and Transportation, Ningbo University, Ningbo 315211, China; yexiaofei@nbu.edu.cn (X.Y.); 18888643916@163.com (Q.Y.); lisong@nbu.edu.cn (S.L.)
2 College of Automobile and Traffic Engineering, Nanjing Forestry University, Nanjing 210037, China; xingchenyan.acad@gmail.com
3 School of Architecture and Transportation, Guilin University of Electronic Technology, Guilin 541004, China
4 National Demonstration Center for Experimental Road and Traffic Engineering Education, School of Transportation, Southeast University, Nanjing 211189, China; chenjun@seu.edu.cn
* Correspondence: wangtao@seu.edu.cn

**Abstract:** The accurate prediction of online car-hailing demand plays an increasingly important role in real-time scheduling and dynamic pricing. Most studies have found that the demand of online car-hailing is highly correlated with both temporal and spatial distributions of journeys. However, the importance of temporal and spatial sequences is not distinguished in the context of seeking to improve prediction, when in actual fact different time series and space sequences have different impacts on the distribution of demand and supply for online car-hailing. In order to accurately predict the short-term demand of online car-hailing in different regions of a city, a combined attention-based LSTM (LSTM + Attention) model for forecasting was constructed by extracting temporal features, spatial features, and weather features. Significantly, an attention mechanism is used to distinguish the time series and space sequences of order data. The order data in Haikou city was collected as the training and testing datasets. Compared with other forecasting models (GBDT, BPNN, RNN, and single LSTM), the results show that the short-term demand forecasting model LSTM + Attention outperforms other models. The results verify that the proposed model can support advanced scheduling and dynamic pricing for online car-hailing.

**Keywords:** online car-hailing; attention mechanism; LSTM; demand forecasting

## 1. Introduction

As an important component of the transport system, on-demand car-hailing and taxi services play increasingly important roles in reducing traffic carbon emissions and easing traffic congestion. However, due to the regulation of the scale of taxi service, passenger demand is far from being met and usually exceeds taxi supply at peak periods. At the same time, the large number of empty vehicles and high operating cost of taxis seriously restrict the development of the taxi market [1]. In 2015, online car-hailing began to gain in popularity in China. Under the promotion of transportation policies, online car-hailing developed rapidly with its high-quality and low-cost travel experience, and its market occupancy continues to increase. By January 2021, the number of users of and orders with online car-hailing services arrived at 500 million and 50 million trips per day, respectively. Compared with the traditional taxi, online car-hailing is more convenient and costs less because of the matches made between passenger and taxi. This real-time connecting concept could also be applied to narrow the gap between supply and demand of taxi trips. However, the uncertain spatial and temporal distributions of online car-hailing and passengers still cause problems, with taxis being in short supply or at a surplus in different periods and areas. The problems of imbalance increase the driving mileage of empty taxis

and passenger waiting time [2]. In order to promote the efficiency of online car-hailing and provide a convenient service for passengers, it is necessary to narrow the gap between supply and demand. The accurate short-term demand forecasting of online car-hailing is one of the effective ways to enable the on-demand platform to understand the spatial and temporal distributions of passengers for enhancing the schedule of ride-hailing cars. It is also helpful for increasing or decreasing the scale of online car-hailing to achieve a reasonable allocation and equilibrium of online car-hailing and passengers.

The demand for online car-hailing is usually relative to the time features, spatial features and weather features of datasets. Most of the existing studies only considered the time features or spatial features of online car-hailing singly, the importance of temporal and spatial sequences are not considered when it comes to increasing prediction performance. Different time series and space sequences have different weights on the demand and supply distributions of online car-hailing. For example, short-term demand in a particular area has a stronger correlation with adjacent regions than with non-adjacent regions. So, too, does temporal distribution of demand; the adjacent time intervals have a higher influence on passenger demand than intervals that are far apart. For example, if passengers during the first 15 min of the peak hour could not take a taxi, they would wait another 15 min for another taxi and add to the demand during the next period. This paper took the historical data of online car-hailing platforms as an example and used the spatial–temporal correlation of online car-hailing demand to develop a short-term demand forecasting model of online car-hailing. In recent years, an attention mechanism has been widely used in various fields. Attention mechanism is used to distinguish the importance of order data between time series and spatial series. In order to better predict the short-term demand for online car-hailing, an attention mechanism was introduced to determine the weights for temporal and spatial sequences. Then, the mixed method combining LSTM and an attention mechanism was applied to forecast the short-term demand of online car-hailing. The main contributions of this paper are as follows: (1) The importance of temporal and spatial sequences for online car-hailing demand are revealed, which provides a basis for extraction scheduling and dynamic pricing of online car-hailing. (2) Based on the historical data of large-scale online car-hailing, a short-term demand forecasting model of online car-hailing is developed, namely, LSTM + Attention. Compared with other models, the accuracy of this model marks a significant improvement.

The rest of the paper is structured as follows. Section 2 presents related work in the area of demand forecasting for online car-hailing. Section 3 details the model structure of LSTM + Attention in detail. Section 4 introduces the results of data analysis. Section 5 introduces the configuration of model parameters. Finally, Sections 6 and 7, respectively, present the application scenarios of online car-hailing demand forecasting and the main conclusions of this work.

## 2. Literature Review

Short-term demand forecasting for online car-hailing belongs to the field of short-term traffic forecasting, which has similar spatial–temporal features. Therefore, short-term traffic forecasting can provide a reference for short-term demand forecasting of online car-hailing. Chen et al. [3] applied LSTM to develop a short-term traffic flow prediction model. The multi-dimensional relative factors of traffic flow and historical traffic data were combined as the input data to achieve high-precision prediction. Zhang et al. [4] proposed a short-term traffic flow forecasting model based on a convolutional neural network (CNN). Li et al. [5] considered the spatial–temporal dependences and weather features as inputs and proposed a regional traffic flow forecasting model through a capsule network and time convolution network (TCN). Ma et al. [6] also used CNN to extract features and predict short-term traffic flow. The results show that the average prediction accuracy of this method is improved by 42.9%, which is better than other methods including random forest, RNN and LSTM. Wang et al. [7] used ConvLSTM to capture temporal and spatial dependencies for a traffic flow demand forecasting model. The prediction results for the

Chengdu DiDi order dataset show that the accuracy and speed of the proposed method is superior to traditional models. Wu et al. [8] used one-dimensional CNN to extract spatial features and two-layer LSTM structure to extract temporal features for forecasting traffic flow. The results show that the model has more advantages than other prediction methods.

Online car-hailing demand is easily disturbed by random factors [9]. Therefore, how to accurately forecast the demand distribution of online car-hailing has become an important research topic in recent years. Xu et al. [10] applied GWO–LSTM to establish a short-term forecasting model of online car-hailing demand, which was verified by actual data and compared with other models. The optimization effect of the proposed model achieved remarkable results compared with the traditional LSTM and BP neural network. Yao et al. [11] proposed a Deep Multi-View Spatial–Temporal Network model, which consists of three parts: LSTM was used to extract temporal correlation, CNN was used to extract spatial correlation, and dynamic time warping was used to calculate the similarity between regions. The results show that this method is superior to other methods. Xu et al. [12] proposed a graph and time series learning model for forecasting taxi demand. The model consisted of a graph attention network and a recurrent neural network. Firstly, the graph attention network was used to learn the spatial dependence, and then the recurrent neural network was used to capture the temporal dependence. Duan et al. [13] combined CNN, LSTM, and ResNet with a deep learning method for taxi demand forecasting. The model used CNN to extract the spatial features, used LSTM to extract the temporal features and integrated this information with external factors (weather, holidays, and air quality index) to predict taxi demand in specific areas. Nejadettehad et al. [14] used three types of recurrent neural networks—simple RNN, GRU, and LSTM with tree-based models (XGBoost and random forest) to predict the short-term demand of car-hailing. The results show that these three types of neural networks are superior to other types of neural networks. Huang et al. [15] proposed a CorrelationNet using Dropconnect method, which added a spatiotemporal correlation analysis mechanism to the deep neural network and formed a new deep learning network CorrelationNet. Then, Dropconnect was used to train CorrelationNet to reduce over fitting. Finally, an application test was carried out in Guangzhou to verify the model. The results show that the method has higher accuracy than other methods. Lu et al. [16] proposed a multi-factor spatiotemporal graph convolution network model which inputs the time dependence and potential spatial dependence of taxi demand, and compares the results with five benchmark models commonly used in traffic flow forecasting. The experimental results show that the model integrating multi-factors could predict taxi demand more accurately. Ke et al. [17] proposed a novel deep multi-task multi-graph learning approach to predict the demand for online car-hailing under different service modes. Independent multi graph revolutionary (MGC) networks for different service modes were established by using the regulated cross task (RCT) learning and the multi-linear relationship (MLR) learning. The results show that the model was obviously better than the benchmark algorithm. Miao et al. [18] proposed a new hierarchical convolution LSTM (HC–LSTM) network, which combined CNN, GCN and LSTM to effectively capture spatiotemporal correlation. In addition, a context net was used to learn context information to help OD prediction. The results show that this method was superior to the existing approaches.

The relevant research emphasized the historical data mining from temporal or spatial features (e.g., [2–4,10]). Although a small number of studies combine the spatiotemporal correlation of online car-hailing features into its demand forecasting model, the demand forecasting of online car-hailing with spatiotemporal features is ignored. In fact, the demand for online car-hailing exhibits a certain temporal and spatial correlation. More importantly, the importances of temporal and spatial sequences are not distinguished for increase the prediction performance (e.g., [11,13]). Different time series and space sequences have different weights on the demand and supply distributions of online car-hailing. On the other hand, although many deep learning methods have been applied in the field of demand forecasting for online car-hailing (e.g., [15–18]), the method of LSTM + Attention

has not been explored in the context of short-term demand forecasting for online car-hailing. An attention mechanism enables recognition the importance of the time series and space sequences of order data. LSTM has higher prediction accuracy and computational efficiency in the existing literature. Therefore, a combined deep learning model of LSTM + Attention is proposed for forecasting short-term online car-hailing demand.

## 3. Methodology

### 3.1. Long Short-Term Memory

LSTM introduces the cell state to avoid the gradient disappearance or gradient expansion of RNN [19]. The cell structure of LSTM is shown in Figure 1.

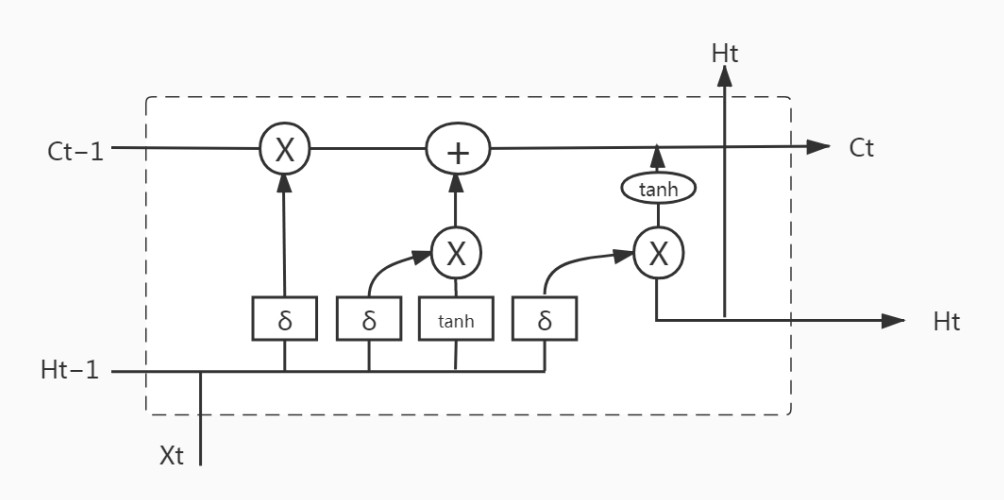

**Figure 1.** Network structure diagram of the LSTM model.

LSTM implements three gate calculations, namely, a forget gate, an input gate and an output gate. The forget gate determines the information that can be abandoned in the cell. The input gate determines the information stored in the cell and the value to be updated. The output gate decides the output content. The calculation formulas for the three gates are expressed as:

$$f_t = \sigma(W_f \times [h_{t-1}, X_t] + b_f) \tag{1}$$

$$i_t = \sigma(W_i \times [h_{t-1}, X_t] + b_i) \tag{2}$$

$$C_t = \tanh(W_c \times [h_{t-1}, X_t] + b_c) \tag{3}$$

$$\tanh(x) = \frac{e^x - e^{-x}}{e^x + e^{-x}} \tag{4}$$

$$C_t = \sigma(f_t \times C_{t-1} + i_t \times C'_t) \tag{5}$$

$$O_t = \sigma(W_o \times [h_{t-1}, X_t] + b_o) \tag{6}$$

$$h_t = O_t \times \tanh(C_t) \tag{7}$$

where $h_{t-1}$ is the output of the previous cell, $X_t$ is the input of this cell, $\sigma$ is the sigmoid function, $W_f$ is the forgetting matrix, $W_i$, $W_c$ and $W_o$ are the weight matrices, $b_i$, $b_c$ and $b_o$ are the deviation vectors, $h_t$ is the output vector of LSTM layer, and $O_t$ represents the output gate.

### 3.2. Attention Mechanism

The importance for demand forecasting of online car-hailing data information at different times is currently insufficiently acknowledged. The model needs to evaluate the importance of online car-hailing data at different times. Thus, the proposed model

introduces the attention mechanism, which considers the different weight coefficients of each input element. It also reasonably selects a small number of key features from a large number of features and gives them more weight to reduce the importance of non-key features, so as to emphasize the influence of key features [20]. The combination of the attention mechanism and LSTM could better emphasize the key features that impact short-term demand forecasting for online car-hailing and increase the forecasting accuracy. The soft-attention method is used in the study, and its structure is shown in Figure 2.

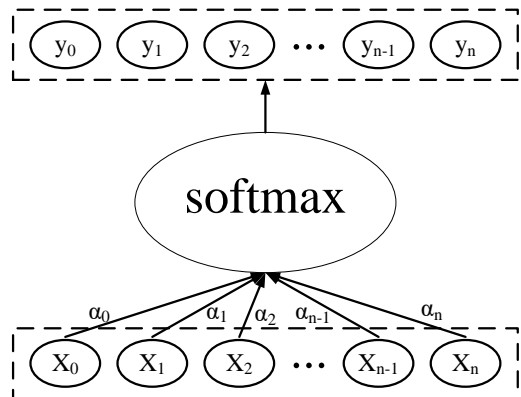

**Figure 2.** Schematic diagram of the attention mechanism.

Among them, $x_n$ represents the input of the model, $\alpha$ represents the attention weight of the attention mechanism on the input data, and $y_n$ represents the output value of the attention mechanism. The formula for calculating the weight coefficient of the attention mechanism can be expressed as follows:

$$v_i^t = v^T \tanh(W_1 x_i + b) \tag{8}$$

$$\alpha_i^t = soft\max(v_i^t) \tag{9}$$

$$y_i^t = \sum_{i=0}^{n} \alpha_i^t x_i \tag{10}$$

### 3.3. LSTM + Attention

A combined attention-based LSTM (LSTM + Attention) model for forecasting short-term demand for online car-hailing was constructed by extracting the temporal features, spatial features and weather features. The flow chart of the model is shown in Figure 3. All the features are input through the input layer, and then are transferred to the attention layer to calculate the attention weight of the features. The output of the attention layer is brought into the LSTM network of three layers in turn, and the final results are obtained through the output layer.

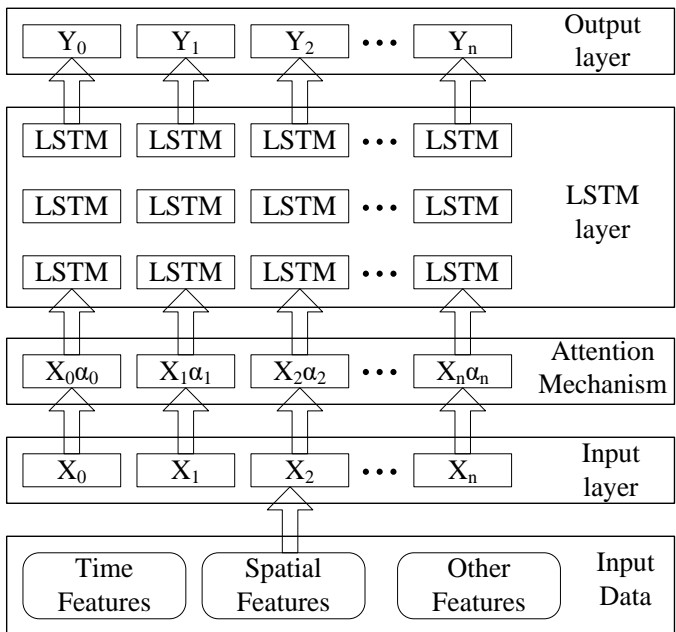

**Figure 3.** LSTM + Attention model flow chart.

## 4. Data Analysis and Results

### 4.1. Time Features

The online car-hailing dataset from 1 May 2017 to 28 May 2017, including working days, weekends and important festivals, was collected. As shown in Figure 4, the demand of online car-hailing obviously changes over the different periods. The demand at 0:00–5:00 on weekdays was very small. It reached the lowest point at 4:00–5:00, and then began to rise. By 7:00 in the morning, the demand of online car-hailing reached the first peak, which was due to the increased commuting demand of students going to school and residents going to work. Subsequently, the demand began to change gently and reached a high demand position at about 4:00 p.m. At 6:00 p.m., the demand reached the maximum. This is due to the increased demand from students after school and residents after work. After this increase, demand began to decline. By about 8:00 p.m., the demand had an upward trend, due to the night life and overtime work. The demand on non-working days was significantly greater than that on working days, and the demand on non-working days rose steadily from 5:00 a.m. to about 6:00 p.m., as residents on non-working days did not need to work and so travel more, resulting in the observed differences. Therefore, this paper extracts the hour attribute information to construct the time segment features.

Figure 5 demonstrates that the demand of online car-hailing changes periodically in a weekly cycle. Therefore, this paper took the week feature as the input feature of a model and uses 0–6 to represent Monday to Sunday respectively. However, as May 1 was a holiday and May 20 was Chinese Valentine's Day, the demand fluctuated greatly, so it is necessary to give special consideration to this information. In this paper, holidays and Valentine's Day were taken as a separate category, with 7 for holidays and 8 for Valentine's Day.

### 4.2. Spatial Features

As shown in Figure 6, Haikou City is divided into a grid of $1 \times 1$ km$^2$. Figure 6 illustrates that the closer passengers are to the city center, the greater is the demand for online car-hailing. Furthermore, correlation analysis results show that the short-term demand in a particular area has a stronger correlation with adjacent regions than with non-adjacent regions. It can be found that the demand for online car-hailing has a certain spatial correlation. This paper takes the adjacent areas as the input features. In addition, some regions that are not adjacent to the target region in space, due to their similarity in land nature, also have certain relevance—that is, there is potential spatial dependence of

online car-hailing demand in different regions. Therefore, a K-means clustering algorithm was applied to conduct the similarity with the traffic time series of the prediction area. A regional demand time series which is similar to the prediction area is obtained for the input model. As shown in Table 1, set K has three values, the area being divided into three categories, in which there are six regions with the same type as the prediction area (Type 2).

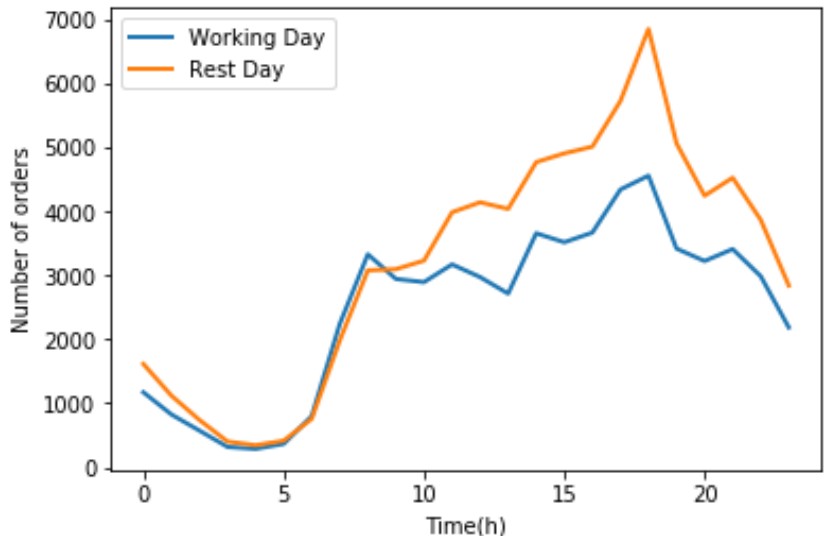

**Figure 4.** Time distribution of online car-hailing travel orders.

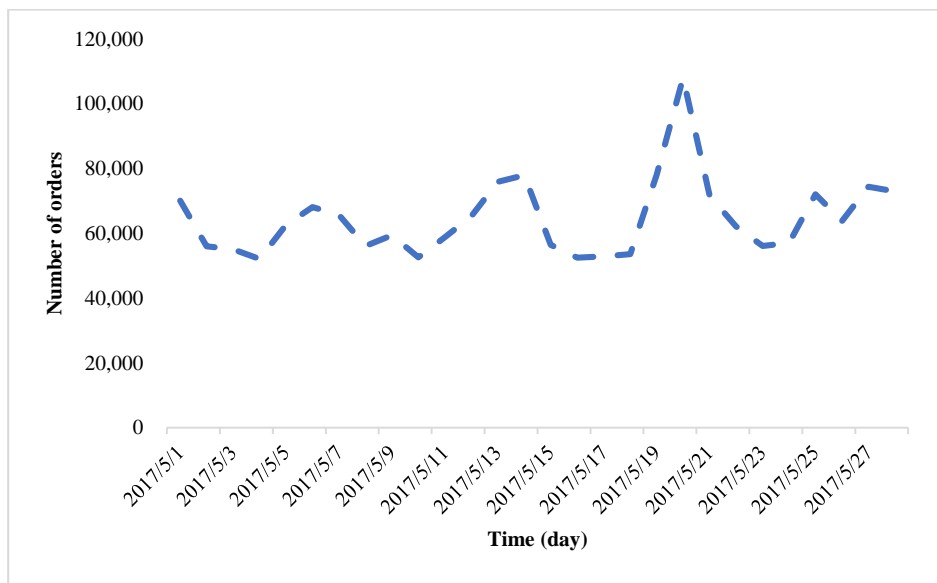

**Figure 5.** Relationships between demand and date of online car-hailing.

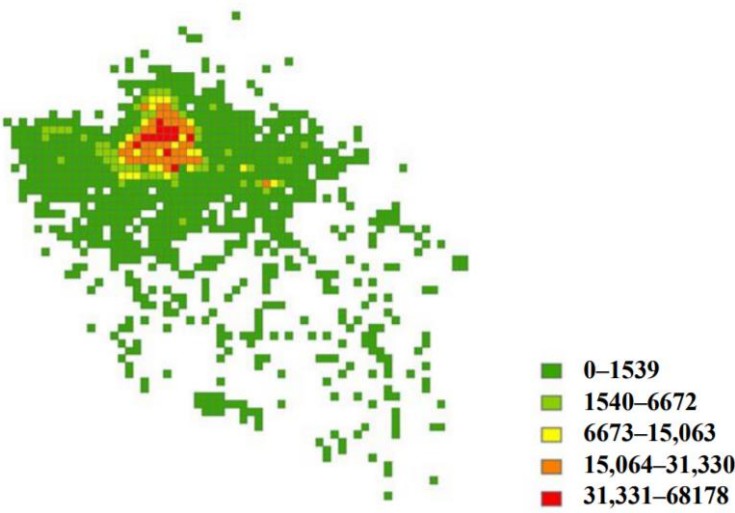

**Figure 6.** Spatial relationship of online car-hailing demand.

**Table 1.** K-means clustering results of similar regions.

| Type | Number of Similar Areas | Proportion |
|------|------------------------|------------|
| Type 1 | 15 | 60% |
| Type 2 | 7 | 28% |
| Type 3 | 3 | 12% |

*4.3. Weather Features*

Different weather conditions lead people to make different travel choices. As shown in Figure 7, cloudy weather occurred on May 9 (Tuesday) and May 11 (Thursday), moderate rain occurred on May 4 (Thursday), and showers occurred on May 16 (Tuesday). The demand for online car-hailing on May 9 and 11 was significantly higher than on May 4 and May 16. Compared with rainy weather, the demand increased significantly (especially at the morning and evening peaks) during cloudy weather. On weekdays, the residents travel more when the weather is good. Therefore, this paper extracted weather information and constructed weather features as input features.

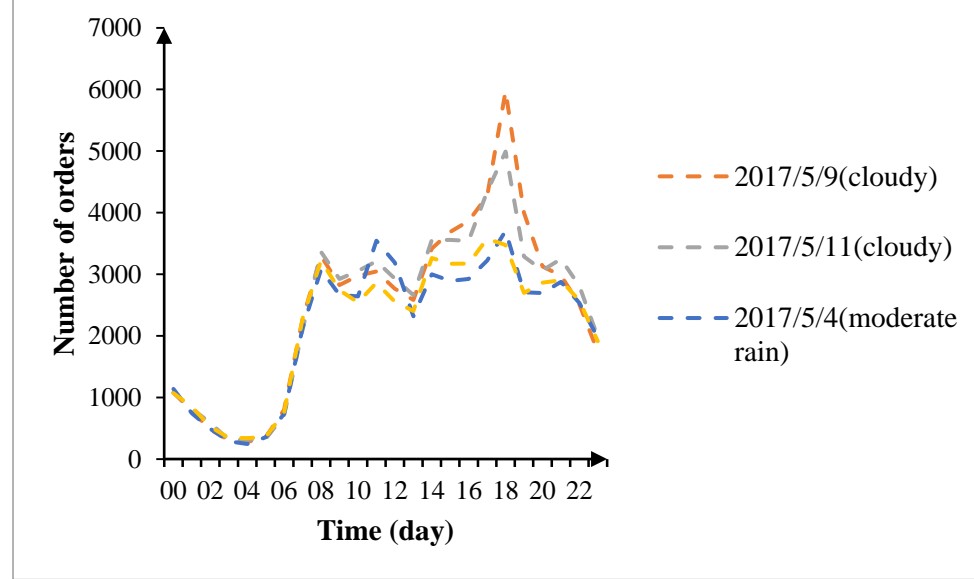

**Figure 7.** Relationship between demand of online car-hailing and weather.

## 5. Case Analysis

### 5.1. Data Source and Preprocessing

The dataset in Haikou City from 1 May to 21 May 2017 was selected as the training set, and the dataset from 22 May to 28 May 2017 was selected as the testing set. Using ArcGIS, Haikou City of China was divided into a 1 km × 1 km grid. As shown in Figure 8, the central area of Haikou City (5 km × 5 km) was taken as the experimental area. Python was used to preprocess the data, remove the redundant data, delete the invalid information, and obtain the demand at different times in each area. The order quantity of each region was counted in 10-min, 15-min and 30-min intervals, respectively. Finally, all the features were counted, the classified variables (week feature, time segment feature, and weather feature) were processed by one-hot encoding, and other features (neighborhood feature and similar area feature) were standardized.

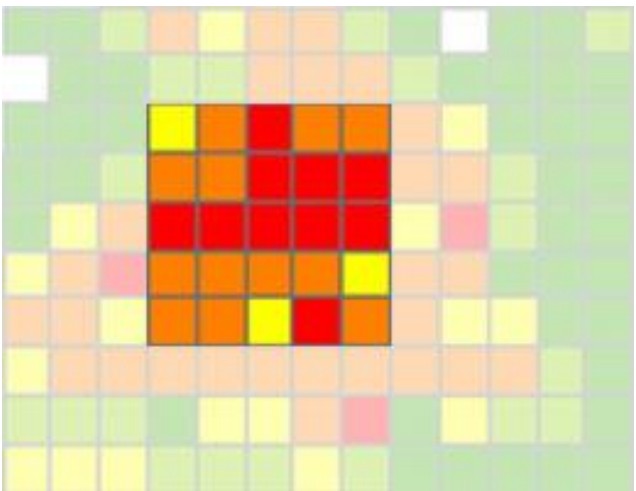

**Figure 8.** Experimental area.

### 5.2. Error Evaluation Index and Parameter Selection

The mean square error (MSE) and the average absolute error (MAE) are introduced to evaluate the performance of the proposed model. The formulas are expressed as:

$$MSE = \frac{1}{n}\sum_{i=1}^{n}\left(f(x_i) - y_i\right)^2 \tag{11}$$

$$MAE = \frac{1}{n}\sum_{i=1}^{n}\left|f(x_i) - y_i\right| \tag{12}$$

In order to improve the effectiveness and performance of the proposed model, the specific parameters are optimized as shown in Table 2.

**Table 2.** Model parameters.

| Parameter Name | 10 min | 15 min | 30 min |
| --- | --- | --- | --- |
| Hidden layer | 3 | 3 | 3 |
| Number of neurons | 64/32/16 | 64/32/16 | 128/64/32 |
| Time step | 1 | 1 | 1 |
| Activation function | relu | relu | relu |
| Optimization function | Adam | Adam | Adam |
| Dropout rate | 0.2 | 0.1 | 0.2 |
| Learning_rate | 0.01 | 0.01 | 0.01 |
| Batch_size | 160 | 150 | 220 |
| Epoch numbers | 100 | 50 | 75 |

### 5.3. Analysis of Predicted Results

In order to verify the effectiveness of the model, four models (LSTM, RNN, BPNN, GBDT) were selected as the comparative analysis. The parameter settings of the deep learning models LSTM, RNN and BPNN were consistent with those in LSTM + Attention. The deep learning model was completed by using the keras module in Python and GBDT was completed by sklearn module. As shown in Table 3 and Figure 9, compared with other models, the short-term demand forecasting model of online car-hailing based on LSTM + Attention decreases in error, with an MSE index of 18.100, 19.949 and 56.854, respectively, and an MAE index of 3.015, 3.472 and 5.680, respectively, both of which are better than the indices of the other four models. In addition, for LSTM + Attention, different time interval divisions have a significant influence on the short-term demand forecasting results for online car-hailing. With the increased time interval for the statistics, the performance of the forecasting model is improved significantly. The reason is that too short a time interval causes an increase in data noise and unhelpful fluctuations. As shown in Figures 10–12, when the time interval is 30 min, the direct fluctuation between the prediction result and the real value is the smallest, and the prediction result is the best. Therefore, the input sequence with 30 min interval has the best prediction performance.

**Table 3.** Error comparison of different models.

| Model | 10 min | | 15 min | | 30 min | |
|---|---|---|---|---|---|---|
| | MSE | MAE | MSE | MAE | MSE | MAE |
| GBDT | 26.805 | 3.848 | 33.661 | 4.256 | 90.131 | 7.1284 |
| BPNN | 24.086 | 3.771 | 25.630 | 3.936 | 71.650 | 6.273 |
| RNN | 22.298 | 3.299 | 24.268 | 3.782 | 64.936 | 6.059 |
| LSTM | 21.841 | 3.3026 | 23.201 | 3.712 | 62.724 | 5.937 |
| LTSM + Attention | 18.100 | 3.015 | 19.949 | 3.472 | 56.854 | 5.680 |

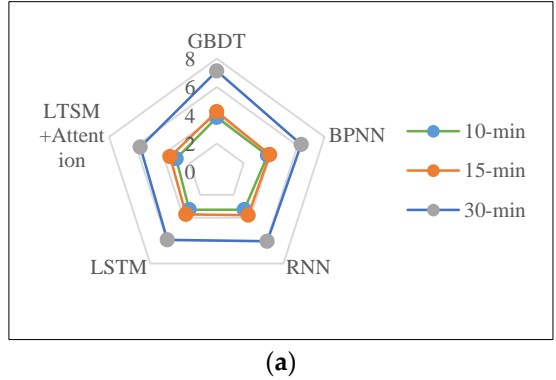

(**a**)

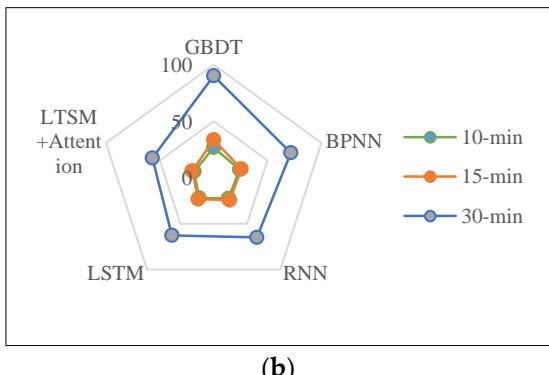

(**b**)

**Figure 9.** Comparison of prediction errors of different models. (**a**) MAE comparison of different models. (**b**) MSE comparison of different models.

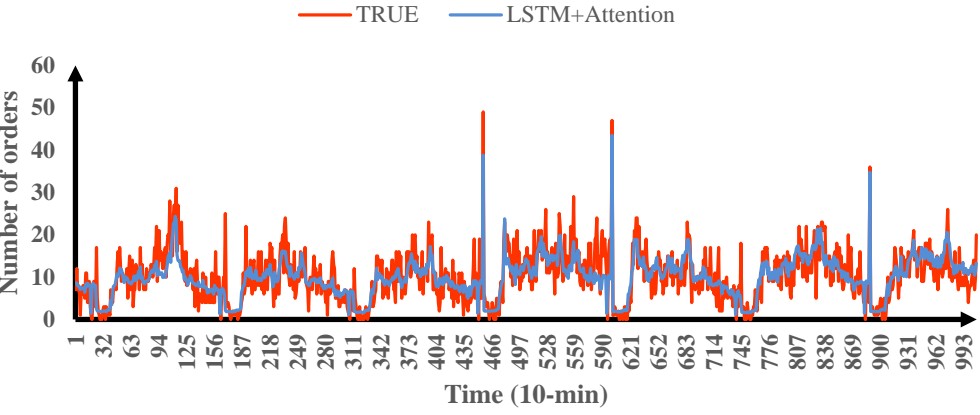

**Figure 10.** Comparison of 10-min prediction values of LSTM + Attention model.

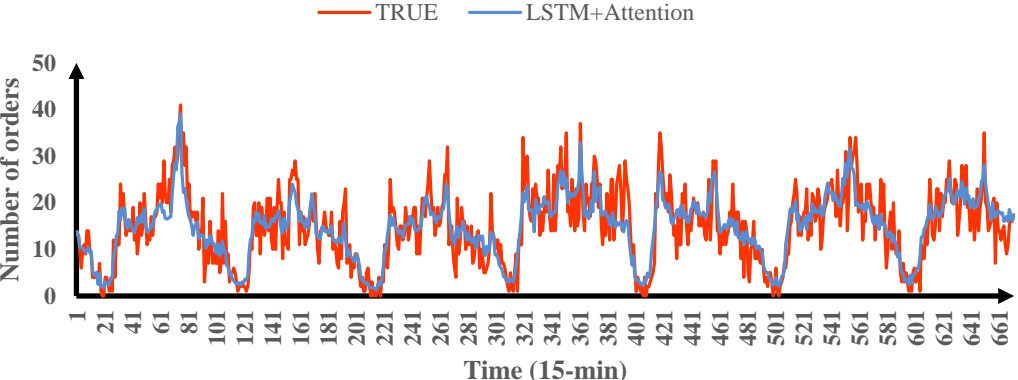

**Figure 11.** Comparison of 15-min prediction values of LSTM + Attention model.

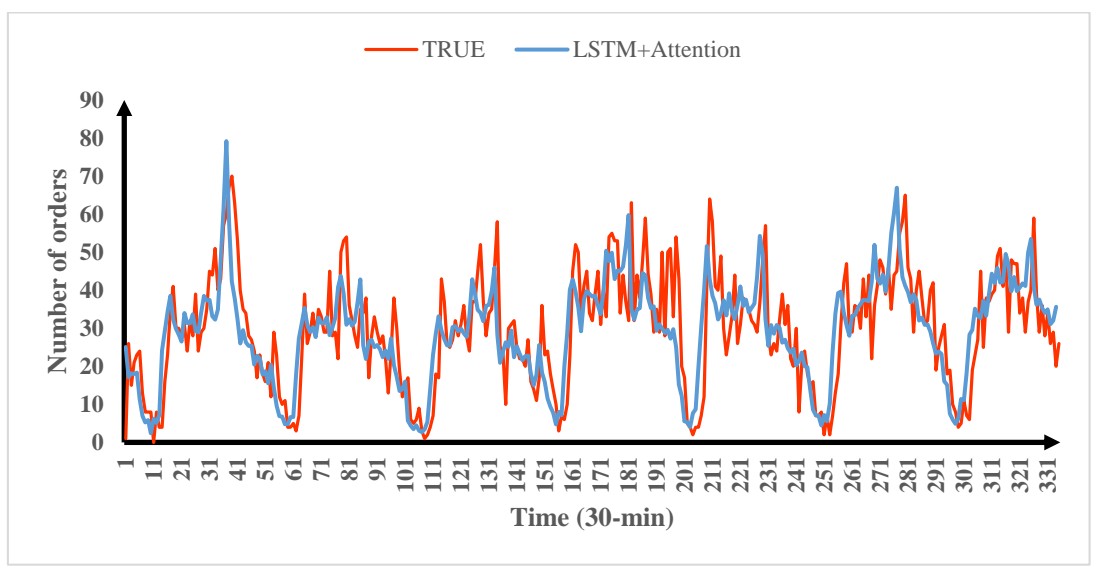

**Figure 12.** Comparison of 30-min prediction values of LSTM + Attention model.

## 6. Application

Short-term demand forecasting has practical significance for the early scheduling and real-time dynamic pricing of online car-hailing services. The supply and demand of online car-hailing could be matched exactly by means of the accurate prediction of demand in the relevant period. Online car-hailing platforms could also advance the scheduling of

car distribution according to car and passenger matching information, so that the regional supply and demand could be balanced, and the traveling time of the empty car and the waiting time of the passenger would be reduced. In addition, given price is an important factor when choosing between online car-hailing services, prices could be altered for specific times and regions according to predicted levels of demand and supply. When the supply is greater than the demand, the price should be reduced. Conversely, when the supply is less than the demand, the price should be increased. Taxis could be directed away from the areas where supply is greater than demand to the areas where supply is less than demand, thereby increasing the number of orders completed.

## 7. Conclusions

This paper has analyzed the factors relevant to the forecasting of online car-hailing demand. In terms of time, the demand for online car-hailing reaches maximum levels in the morning and evening peaks, while the demand for online car-hailing on rest days is much greater than on working days. The demand for online car-hailing shows a certain periodic change law according to time. As regards space, a K-means clustering algorithm has been used to find similar regions and adjacent regions as spatial features. In addition, the online car-hailing orders under different weather conditions have been analyzed to verify that weather characteristics have a certain impact on online car-hailing demand forecasts.

In order to accurately forecast the short-term demand of online car-hailing in different regions of the city, a combined attention-based LSTM (LSTM + Attention) model was constructed by extracting temporal features, spatial features and weather features. Significantly, the attention mechanism enables the distinguishment of the time series and space sequences of order data. A small number of key features were reasonably selected from a large number of features and given greater weight. The combination of an attention mechanism and LSTM could better emphasize the key features that impact the short-term demand forecasting of online car-hailing and increase forecasting accuracy.

The Attention Mechanism was combined with LSTM to generate better predictions. Finally, the order data in Haikou city was collected as the training and testing datasets. The data were divided into 10-min, 15-min, and 30-min intervals for predictions. Compared with other prediction models (GBDT, BPNN, RNN, and LSTM), the results show that the LSTM + Attention model is superior to other models. At the same time, the attention mechanism is applied to effectively allocate weights to distinguish the importance of time and space sequences, which helpfully increases the training speed of the model and improves the computational efficiency of the model. When the time interval is 30 min, the direct fluctuation between the forecasting results and real values is smallest, and the forecasting result is the best. MSE and MAE were improved by 9.36% and 4.33%, respectively, which verified the predictive performance of the model.

Although this paper considered time features, spatial features, and weather features as input features, the short-term demand of online car-hailing is also related to other factors. Further work will consider the complex road network correlation and congestion index. In addition, due to limited data, this paper only validates the model on the data set of Haikou, China. In future work, we will further verify the applicability of the model on other data sets.

**Author Contributions:** Methodology, X.Y. (Xiaofei Ye); data curation, Q.Y. and S.L.; model, X.Y. (Xiaofei Ye) and Q.Y.; validation, T.W. and X.Y. (Xingchen Yan); writing—original draft preparation, X.Y. (Xiaofei Ye), Q.Y. and T.W.; writing—review and editing, X.Y. (Xiaofei Ye) and Q.Y.; supervision, J.C.; project administration, X.Y. (Xiaofei Ye); funding acquisition, T.W. All authors have read and agreed to the published version of the manuscript.

**Funding:** This research was supported by the Natural Science Foundation of Zhejiang Province, China (No.LY20E080011 & LY21E080008), the National Natural Science Foundation of China (No. 71971059, 71701108 & 71861006), Guangxi Natural Science Foundation (No. 2020GXNSFAA159153), Guangxi Science and Technology Base and Talent Special Project (AD20159035), Natural Science Foundation of Jiangsu Province (grant no. BK20180775), the Basic Public Welfare Research Project of Zhejiang Province 2018 (LGF18E090005) and the National Key Research and Development Program of China—Traffic Modeling, Surveillance and Control with Connected and Automated Vehicles (2017YFE9134700).

**Data Availability Statement:** The data used to support the findings of this study are available from the corresponding author upon request.

**Acknowledgments:** Data source: Didi Chuxing GAIA Initiative. The authors thank their mentor, Xiaofei Ye of the Ningbo University, who gave instruction on writing this paper. The authors also thank the interviewers for their assistance in the survey.

**Conflicts of Interest:** The authors declare that they have no conflicts of interest.

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
