# Peer review of "Demand Forecasting of Online Car-Hailing with Combining LSTM + Attention Approaches"

_electronics, doi:10.3390/electronics10202480_

Round 1

Reviewer 1 Report

In this manuscript, authors proposed a combined Attention-based LSTM model for forecasting online car-hailing short-term demand, considering temporal features, spatial features, and weather features. Even though the readers has enough interested in the proposed system, some questions need to be solved for publishing as follows:

  1. Demand for forecasting online car-hailing can change according to regional and time-specific issues. However, authors can clearly mention weathers issues whether time characteristic is considered or not. In this manuscript, authors may together explain impacts on the day based weather change to Table 3, Figure 9, and Section 4.3. If possible, attention-based LSTM may be considered with weather and without weather conditions
  2. Please mention the explanation of the short-term period of the demand in detail. If the models have bounds below the 30 minutes, can we call as the short-term period? The allowable bound for the short-term period should be defined.
  3. In Literature Review (Section 2), please mention differences on the previous work (reference [2]).
  4. Editorial issues to be updated are included a lot.

Author Response

Response to Reviewer 1 Comments

First of all, I would like to thank the editor for arranging the review and the reviewer for your valuable suggestions. The authors have carefully answered the questions point-by-point in accordance with the requirements of yours, and made careful modifications to the article, and all of revisions have been clearly highlighted. Because of your suggestions, the revised articles become better and readers can get more valuable information. Thanks again for your help.

Point 1: Demand for forecasting online car-hailing can change according to regional and time-specific issues. However, authors can clearly mention weathers issues whether time characteristic is considered or not. In this manuscript, authors may together explain impacts on the day based weather change to Table 3, Figure 9, and Section 4.3. If possible, attention-based LSTM may be considered with weather and without weather conditions

Response 1:In the experiment, we consider the impact of deleting one of the features on the prediction results, and conduct a comparative experiment. The results show that the prediction error increases after deleting the weather features.

Point 2: Please mention the explanation of the short-term period of the demand in detail. If the models have bounds below the 30 minutes, can we call as the short-term period? The allowable bound for the short-term period should be defined.

Response 2: After reading other literatures, we believe that when the prediction time is less than 30-min, it is short-term demand prediction

Point 3: In Literature Review (Section 2), please mention differences on the previous work (reference [2]).

Response 3:We illustrate this problem in Section 2. Compared with other work, we propose an online car Hailing demand forecasting method based on LSTM + attention, and consider the impact of time features, spatial features and weather features on online car-hailing demand forecasting. And use Attention mechanism emphasized on the distinguishing the importance of time series and space sequences of order data.

Reviewer 2 Report

The article as a whole is well-established, and the subject is interesting. The following comments may assist in improving the article:

  • The authors should ask the help of native English-speaking proofreader, because there are some minor linguistic mistakes that should be fixed.
  • More suitable title should be selected for the article.
  • It is proposed that the article be supplemented with a flowchart illustrating the research technique.
  • A review of the literature is insufficient. It is critical to include some recent work (2018–2020) in the review of the literature.
  • Some further explanations and interpretations are required for the results.
  • It is recommended to include a discussion of the findings with additional explanation/details and a conclusion with future work.

Author Response

Response to Reviewer 2 Comments

First of all, I would like to thank the editor for arranging the review and the reviewer for your valuable suggestions. The authors have carefully answered the questions point-by-point in accordance with the requirements of yours, and made careful modifications to the article, and all of revisions have been clearly highlighted. Because of your suggestions, the revised articles become better and readers can get more valuable information. Thanks again for your help.

Point 1: The authors should ask the help of native English-speaking proofreader, because there are some minor linguistic mistakes that should be fixed.

Response 1:We corrected the linguistic of the article.

Point 2:More suitable title should be selected for the article.

Response 2:The main work of this paper is to use the LSTM + Attention method to predict the online car-hailing demand. We think the title of the article can well summarize the main content of the article.

Point 3:It is proposed that the article be supplemented with a flowchart illustrating the research technique.

Response 3:We believe that figure 3 has explained the prediction process of the model in detail. In Figure 3, we illustrate the input characteristics , the structure and the output of the model in detail.

Point 4:A review of the literature is insufficient. It is critical to include some recent work (2018–2020) in the review of the literature.

Response 4:We have revised this problem and added a literature review of recent three years in Section 2.

Point 5:Some further explanations and interpretations are required for the results.

It is recommended to include a discussion of the findings with additional explanation/details and a conclusion with future work.

Response 5:We revised the conclusion and added the discussion of the analysis results. “This paper analyzes the influencing factors of online car-hailing demand forecast. In terms of time, the demand for online car-hailing reaches the maximum in the morning and evening peak, while the demand for online car-hailing on rest days is much greater than that on working days. The demand for online car-hailing shows a certain periodic change law with time. In space, this paper uses K-means clustering algorithm to find similar regions and adjacent regions as spatial features. In addition, the online car-hailing orders under different weather conditions are analyzed to verify that the weather characteristics have a certain impact on the online car-hailing demand forecast.”

Reviewer 3 Report

Thank you for providing me an opportunity to participate in the revision process. The manuscript addresses the problem of forecasting the demand for online car-hailing. Specifically, in this work, the authors utilized three types of features (i.e., time features, spatial features, and weather features) as inputs to a combined LSTM+Attention Mechanism model to achieve the predicted values. The experimental results demonstrated the improved accuracy of prediction in comparison with some of the conventional deep learning-based models. There are some comments listed below that are to be addressed prior to considering the manuscript for publication. Specifically:

1) Paragraph describing the organization of the manuscript should be added to the introduction section for helping the reader to navigate through the paper.

2) Please, use the same notations in Eq. (1-7) and Figure 1 to improve the readability of this part of the manuscript.

3) It is recommended to modify Figure 2 to better reflect the ideas presented in Eq. (8-10). It can make it easier for inexperienced readers to better understand the implementation of the Attention Mechanism in this work.

4) It can be useful if the authors highlight the specific time-points in Figure 4 that were described in Section 4.1 to help readers navigating through this figure.

5) According to Table 2, the authors use different model configurations for different prediction intervals. Is it possible to use the same model configuration for 3 different intervals? How the performance would be in this case? I am curious, is it really necessary to utilize different model configurations in these study cases?

6) The results presented in Table 3 are promising. Is it possible to add comparisons with some of the state-of-the-art approaches published in the literature that address the same domain problem (i.e., forecasting of car-hailing demand)? 

7) How did the authors determine the number of epochs used for training the models? Was some sort of early stopping approach used to determine this? It can be useful for the readers to understand the criteria used for stopping the training process.

8) Was the proposed model trained in the "online" manner (the training data was inputted batch-by-batch and the predictions were made immediately) or in an "offline" manner (i.e., first, the model was trained on the whole training data and then the predictions were made)?

9) There are typos in the caption of Figure 12. Please, consider fixing this.

Author Response

Response to Reviewer 3 Comments

First of all, I would like to thank the editor for arranging the review and the reviewer for your valuable suggestions. The authors have carefully answered the questions point-by-point in accordance with the requirements of yours, and made careful modifications to the article, and all of revisions have been clearly highlighted. Because of your suggestions, the revised articles become better and readers can get more valuable information. Thanks again for your help.

Point 1: Paragraph describing the organization of the manuscript should be added to the introduction section for helping the reader to navigate through the paper.

Response 1:This section has been added in the introduction section “The rest of the paper is structured as follows. Section 2 presents related work in the area of demand forecasting of online car-hailing. Section 3 details the model struc-ture of LSTM and attention in detail. Section 4 introduces the results of data analysis. Section 5 introduces the configuration of model parameters. Finally, sections 6 and 7 respectively present the application scenarios of online car-hailing demand forecasting and the main conclusions of this work.”

Point 2: Please, use the same notations in Eq. (1-7) and Figure 1 to improve the readability of this part of the manuscript.

Response 2:We have added the interpretation of Eq. (1-7) notations.

Point 3: It is recommended to modify Figure 2 to better reflect the ideas presented in Eq. (8-10). It can make it easier for inexperienced readers to better understand the implementation of the Attention Mechanism in this work.

Response 3:Attention mechanism considers the different weight coefficients of each input element. It also reasonably se-lects a small number of key features from a large number of features and gives them more weights to reduce the importance of non-key features.In Figure 2,  represents the input of the model,  represents the attention weight of the Attention Mechanism on the input data, and  represents the output value of the Attention Mechanism. Eq(8-9) shows that each feature is given different weights, and formula Eq(10) shows the output of the model.

Point 4: It can be useful if the authors highlight the specific time-points in Figure 4 that were described in Section 4.1 to help readers navigating through this figure.

Response 4:Section 4.1 is modified and the description of data time is added. “The online car-hailing data set from May 1, 2017 to May 28, 2017 including working days, weekends and important festivals were collected.”

Point 5: According to Table 2, the authors use different model configurations for different prediction intervals. Is it possible to use the same model configuration for 3 different intervals? How the performance would be in this case? I am curious, is it really necessary to utilize different model configurations in these study cases?

Response 5: In the experiment, we considered whether the same parameters can be used, and carried out the experiment. The results show that when the parameters are the same, the prediction results are not as good as when the parameters are set separately.

Point 6: The results presented in Table 3 are promising. Is it possible to add comparisons with some of the state-of-the-art approaches published in the literature that address the same domain problem (i.e., forecasting of car-hailing demand)?

Response 6:Because the data sources of similar problems are quite different, most prediction methods are data dependent, and the results of mutual comparison are not comparable.

Point 7: How did the authors determine the number of epochs used for training the models? Was some sort of early stopping approach used to determine this? It can be useful for the readers to understand the criteria used for stopping the training process.

Response 7:We determine the number of epochs used for training the models hrough a large number of experiments, compared with some sort of early stopping approach, this approach takes more calculation and time, but it can also get the same result.

Point 8: Was the proposed model trained in the "online" manner (the training data was inputted batch-by-batch and the predictions were made immediately) or in an "offline" manner (i.e., first, the model was trained on the whole training data and then the predictions were made)?

Response 8:We use an "offline" manner ( the model was trained on the whole training data and then the predictions were made).

Point 9: There are typos in the caption of Figure 12. Please, consider fixing this.

Response 9: We corrected it. “Figure 12. Comparison of 30-minute prediction values of LSTM +Attention model”

Round 2

Reviewer 1 Report

In Point 1, my comments was that the authors did not consider the weather variation during the day. Also, on adding the results for with and without weather conditions, authors mentioned it only in the reply letter. I cannot find any clue in the revised manuscript.

In Point 2, the authors did not define why the short term period is set to 30 min with the proper reference.

In Point 3, the authors clearly should add the difference in their previous work (e.g. reference [2]) in Section 2 (Literature Review).

Finally, any editorial updates were not found in the revised manuscript. Authors omitted the last comment.

Thus, I cannot allow this manuscript to be published.  

my comment file is attached additionally.

Author Response

Response to Reviewer 1 Comments

First of all, I would like to thank the editor for arranging the review and the reviewer for your valuable suggestions. The authors have carefully answered the questions point-by-point in accordance with the requirements of yours, and made careful modifications to the article, and all of revisions have been clearly highlighted. Because of your suggestions, the revised articles become better and readers can get more valuable information. Thanks again for your help.

Point 1: In Point 1, my comments was that the authors did not consider the weather variation during the day. Also, on adding the results for with and without weather conditions, authors mentioned it only in the reply letter. I cannot find any clue in the revised manuscript.

Response 1:Due to the data, this paper does not consider the weather change in a day. In addition, due to space reasons, the prediction results of deleted weather features are not included in the original. As shown in Table 1, the prediction results of the model increase after deleting weather features.

Table 1 Model prediction comparison

Model

10min

15min

30min

MSE

MAE

MSE

MAE

MSE

MAE

LTSM+Attention

(without weather conditions)

20.198

3.227

21.058

3.522

58.052

5.834

LTSM+Attention

(with weather conditions)

18.100

3.015

19.949

3.472

56.854

5.680

Point 2: In Point 2, the authors did not define why the short term period is set to 30 min with the proper reference.

Response 2: First, we find that many literatures use 30 minutes as the prediction cycle. Second, the prediction interval is too long to be used for online car-hailing dynamic pricing and real-time scheduling; It is too short and the data fluctuates greatly, which does not conform to the daily travel law of online car-hailing, so we use the 30 minute interval adopted by everyone as the statistical interval.

Point 3: In Point 3, the authors clearly should add the difference in their previous work (e.g. reference [2]) in Section 2 (Literature Review). (reference [2]).

Response 3:We modified section 2 to add the difference from the previous work. “The relevant researches emphasized the historical data mining from temporal or spatial features (e.g. reference [2-4,10]). Although a small number of studies combine the spatiotemporal correlation of online car-hailing features into its demand forecast-ing model, the demand forecasting of online car-hailing with spatiotemporal features is ignored. In fact, the demand for online car-hailing has a certain temporal and spatial correlation. More importantly, the importances of temporal and spatial sequences are not distinguished for increase the prediction performance (e.g. reference [11,13]). Ac-tually, different time series and space sequences have different weights on the demand and supply distributions of online car-hailing. On the other hand, although many deep learning methods have been applied in the field of demand forecasting of online car-hailing (e.g. reference [15-18])., LSTM+Attention method has not been explored in the short-term demand forecasting of online car-hailing. Attention mechanism em-phasized on the distinguishing the importance of time series and space sequences of order data. LSTM has higher prediction accuracy and computational efficiency in the existing literature. Therefore, a combined deep learning model with LSTM+Attention is proposed for forecasting short-term online car-hailing demand.”

Point 4: Finally, any editorial updates were not found in the revised manuscript. Authors omitted the last comment.

Response 4:We modified the grammar and format of the manuscript.

Reviewer 3 Report

The manuscript has been improved compared to the previous version along with some clarifications added. As a reviewer, I do not have further comments,

Author Response

thanks for your attention

Round 3

Reviewer 1 Report

Kindly, auhors presented the proper answers for reviewer's question.

Thank you for replying to the second request.

I'd like to recommend this revised manuscript to the publication.